# Divide and Rule: Phase Separation in Eukaryotic Genome Functioning

**DOI:** 10.3390/cells9112480

**Published:** 2020-11-15

**Authors:** Sergey V. Razin, Sergey V. Ulianov

**Affiliations:** 1Institute of Gene Biology, Russian Academy of Sciences, 119017 Moscow, Russia; sergey.v.razin@usa.net; 2Faculty of Biology, M.V. Lomonosov Moscow State University, 119017 Moscow, Russia

**Keywords:** liquid-liquid phase separation, LLPS, chromatin spatial organization, enhancer-promoter communication

## Abstract

The functioning of a cell at various organizational levels is determined by the interactions between macromolecules that promote cellular organelle formation and orchestrate metabolic pathways via the control of enzymatic activities. Although highly specific and relatively stable protein-protein, protein-DNA, and protein-RNA interactions are traditionally suggested as the drivers for cellular function realization, recent advances in the discovery of weak multivalent interactions have uncovered the role of so-called macromolecule condensates. These structures, which are highly divergent in size, composition, function, and cellular localization are predominantly formed by liquid-liquid phase separation (LLPS): a physical-chemical process where an initially homogenous solution turns into two distinct phases, one of which contains the major portion of the dissolved macromolecules and the other one containing the solvent. In a living cell, LLPS drives the formation of membrane-less organelles such as the nucleolus, nuclear bodies, and viral replication factories and facilitates the assembly of complex macromolecule aggregates possessing regulatory, structural, and enzymatic functions. Here, we discuss the role of LLPS in the spatial organization of eukaryotic chromatin and regulation of gene expression in normal and pathological conditions.

## 1. Introduction

The eukaryotic cell nucleus contains several functional compartments, such as the nucleolus, speckles, Cajal bodies, and PML bodies [1]. These compartments are not surrounded by membranes but yet are clearly distinct from the rest of the nucleoplasm. The mechanisms of nuclear compartments assembly have been discussed over many years [2,3,4,5]. Another actively discussed issue concerns the relationship between cell nucleus functional compartmentalization and genome spatial organization [1,6,7]. Recent results demonstrate that liquid-liquid phase separation (LLPS) plays an important role in both the functional compartmentalization of the eukaryotic cell nucleus and 3D genome organization [8]. LLPS is a process involving components in a solution separating into two or more distinct phases with different physical and chemical properties [9] demarcated with a well-defined surface kinetically and thermodynamically constraining a free diffusion of the phase components [10]. This process can be illustrated by the formation of oil droplets in an aqueous solution. In a cell nucleus, LLPS is driven by multiple and relatively nonspecific biopolymer interactions. In particular, proteins possessing intrinsically disordered domains (IDRs) are known to form liquid droplets via LLPS [11]. Many nuclear proteins possess such IDRs and, hence, are capable of forming phase-separated condensates in the overcrowded nuclear milieu [12,13,14,15,16,17,18,19]. Although in biological systems, the hydrophobic interactions between IDRs are considered to be the main driving force of LLPS, the electrostatic interaction between charged residues in IDRs and between proteins and nuclear acids may contribute as well [20,21].

LLPS occurs when the concentration of interacting molecules reaches a threshold value, which can be achieved within a local neighborhood due to a gathering of interacting molecules on a platform possessing specific affinity sites for macromolecules capable of forming phase condensates [21]. This process is typically referred to as seeding nucleation of condensate formation [9]. The organizing platform may be a protein [22,23], RNA [24,25], polyADP-ribose [26], DNA [27,28], or chromatin fibril [28]. The phase condensates generated by LLPS are expected to possess a round shape, fuse upon coalescence, and quickly exchange components with the external milieu [29,30]. To this end, it should be mentioned that a rapid exchange with the nucleoplasm is typical for proteins deposited in various nuclear compartments [31,32]. Although LLPS currently forms the main focus of most research of nuclear compartmentalization, it should be mentioned that long polymers, such as chromatin fiber, may be compartmentalized by another process termed polymer-polymer phase separation (PPPS), which can be triggered by introducing links bridging different sections of the polymer [33]. PPPS can be regarded as a coil-globule transition [34,35,36]. The distinctive feature of PPPS is that the bridges themselves are essential irrespective of the nature of the bridging molecules [33]. These bridging molecules are not supposed to establish multiple interactions between themselves or form phase condensates in the absence of a polymer even at high bridging molecule concentrations [33]. Consequently, the solvent inside the collapsed polymer globule should not necessarily be phase-separated from the external milieu [33]. Below, we will discuss how various phase separation mechanisms contribute to eukaryotic cell nucleus compartmentalization and 3D organization of the genome. The role of phase separation in the assembly of nuclear bodies has been addressed in several recent reviews [37,38,39]. Therefore, here we focus on the recent data demonstrating the contribution of LLPS and PPPS in spatial genome organization, assembly of active and repressed chromatin compartments, and transcription control. We discuss the current model of enhancer action that postulates an LLPS-driven assembly of activating compartments on enhancers and superenhancers. Finally, we consider human pathologies caused by LLPS deregulation.

## 2. Global Organization of the Cell Nucleus

LLPS first drew attention in connection with the assembly of the so-called nuclear bodies, such as the nucleolus, nuclear speckles, Cajal bodies, and ND10 bodies [5,38]. Nuclear bodies possess all of the expected characteristics of phase-separated condensates. They are not surrounded by membranes, have a spherical shape, and comprise a number of proteins that are rapidly exchanged with the nucleoplasmic pull. Furthermore, nuclear bodies can both disassemble and fuse [1]. The sets of proteins present in various nuclear bodies can partially overlap but are still specific [39]. This specificity is imposed by the nucleation of liquid condensate assembly at a specific interaction platform, which may be non-coding RNA as in paraspeckles [40] or protein as in PML-bodies [41].

The role of LLPS in global nuclear space organization is likely not restricted to the assembly of nuclear bodies. The recent model of eukaryotic cell nucleus organization discriminates the chromatin domain represented by chromosomal territories and the interchromatin compartment (IC) [42,43]. The physical reason for the existence of channels between chromosomal territories and within chromosomal territories remains elusive. It is tempting to suggest that chromatin and IC also form separated phases (Figure 1). To this end, it is of note that under physiological conditions, chromatin can form phase-separated droplets due to interactions mediated by the histone tails [44]. The perichromatin layer lining the interchromatin channels is represented by transcriptionally active chromatin [42,43], which possesses different properties compared to bulk chromatin due to a high level of histone acetylation preventing internucleosomal interactions. This transcriptionally active chromatin may even form a distinct liquid condensate due to the recruitment of multi-bromodomain proteins [44]. Finally, interchromatin channels should be filled with RNA and RNA-binding proteins, which both possess the ability to interact, triggering LLPS [45,46]. Nuclear bodies assembled within the IC [42,43] may constitute a physical obstacle for the coalescence and fusion of neighboring chromatin masses [47].

## 3. 3D Genome

### 3.1. Current View of 3D Genome Organization

3D genome organization has become a hot topic in molecular biology because it has been demonstrated that for transcription activation remote enhancers should establish spatial contact with target promoters, a process that is only possible at the 3D genome level [48,49,50]. In addition, it has long been known that active and repressed genomic regions, commonly referred to as euchromatin and heterochromatin, are folded in a different manner [51]. The modern view of 3D genome organization is mainly based on the results obtained using high-throughput chromosome conformational capture (Hi-C) analysis [52]. At low resolution, this analysis demonstrated that active and repressed chromatin domains are spatially segregated within the so-called A and B compartments [52]. The chromatin chain in both A and B compartments is folded into self-interacting domains termed “topologically associating domains” (TADs) [53,54,55], which may coincide with chromatin loops or harbor several of such loops [56,57]. Analysis performed at higher resolution demonstrated that TADs and compartmental domains coexist at the same genomic scale [58,59]. Whereas there are several lines of evidence that TADs are assembled via active DNA loop extrusion [60,61,62,63], compartmental domains appear to form by condensation of nucleosomes bearing particular epigenetic marks [58,59]. Here, LLPS comes to a stage [33]. Both histones and many non-histone proteins associated with chromatin possess IDRs and under certain conditions are capable of forming either liquid condensates or gels [15,16,17,28,44,64,65] compacting chromatin into globular structures manifested as self-interacting domains in Hi-C maps.

Chromatin fiber can be regarded as a polymer comprising alternating blocks of different nature (in the simplest case of euchromatic and heterochromatic regions, although both euchromatin and heterochromatin may be further divided into several subclasses [57,66]). Under certain conditions, block copolymers of this type undergo microphase separation resulting in clustering of the blocks of similar types. The parameters of clusters (including the average number of blocks present in a single cluster) will depend on several conditions, including the size of blocks and their mutual affinity (the ability to establish links between blocks of the same type) [67]. Below, we shall discuss how LLPS and PPPS contribute to the spatial segregation of repressed and active chromatin.

### 3.2. Role of Phase Separation in Heterochromatin Domain Assembly

Several observations show that LLPS is essential for the assembly of both constitutive and facultative heterochromatin. It has long been assumed that the dense packaging of DNA in heterochromatin physically prevents DNA accessibility to transcription factors and components of transcription machinery [68,69]. Recent data suggest that this is not the case because relatively large molecules can easily permeate through both euchromatic and heterochromatic regions [70].

Furthermore, some essential genes are located in heterochromatin and are transcribed [71]. It thus appears that heterochromatin constitute only a distinct chromatin compartment in which some proteins remain, whereas the others either do not enter it or are not retained. Several observations suggest that this compartment is formed via LLPS. Thus, structural components of both constitutive (HP1) and facultative (Polycomb) heterochromatin possess IDRs and can trigger LLPS in vitro and within living cells [16,17,38,64,72]. Of note, histone H1, which is overrepresented in repressed chromatin, condenses into liquid-like droplets in the nuclei of living cells. The nuclear foci of H1 possess some expected features of LLPS-derived condensates and colocalize with HP1α and dense DNA of heterochromatin domains [28]. In vitro, H1 alone does not form droplets but does form them in the presence of DNA or nucleosomes. H1 also can form phase-separated condensates with polynucleosomes, but these condensates have an irregular shape that differentiates them from the classical droplets generated by LLPS [28]. It should be mentioned, however, that conditions in the cell nucleus differ substantially from those in vitro. In particular, in the above-described experiments, the level of molecular crowding was not comparable to that within the cell nucleus. Further, the presence of high amounts of various RNA molecules can seriously affect protein interactions. Hence, both the ability and inability of a specific protein to form liquid condensates in vitro should be treated with caution when the potential contribution of this protein in the formation of liquid condensates within the cell nucleus is considered.

Another nuclear protein that is likely to contribute to LLPS in the pericentromeric chromatin is Scaffold Associated Factor B (SAFB), previously known as nuclear matrix protein [73]. SAFB interacts with heterochromatin-associated repeat transcripts and promotes phase separation [74].

Although all of the above-discussed observations suggest the role of LLPS in heterochromatin formation, the typical heterochromatin domains possess some features that distinguish them from liquid condensates. These domains do not necessarily have a round shape, and their crucial components, such as HP1, do not rapidly exchange with the nucleoplasmic pull. It is, therefore, possible that after being initially assembled, the LLPS heterochromatic domains eventually undergo gelation [64,75]. Gelation is a transition from a solution of dispersed monomers and oligomers to a system-spanning network [76]. This process, termed liquid–gel phase separation (LGPS), differs from PPPS described below (see [77]). Gelation driven by phase separation requires lower protein concentrations and seems to be quite common in biological systems [76]. Besides being able to phase separate under certain conditions, HP1 is also known to bridge nucleosomes bearing H3K9me3 epigenetic marks [78,79]. There are many other architectural proteins that can bridge remote parts of a chromatin fiber [80,81]. Electrostatic interactions between nucleosomes mediated by histone tails also contribute to establishing links within chromatin fiber [82]. Establishing multiple links between distinct regions of a chromatin fiber should cause polymer-polymer phase separation (PPPS) (Figure 2) [33]. In accordance with this supposition, the results of a recent study of mouse heterochromatin suggest that it does not possess the expected properties of LLPS-derived condensates [83]. Furthermore, analysis of mechanistic properties of heterochromatin suggests that heterochromatin is solid rather than liquid [84,85]. It should be mentioned that a contribution of PPPS and LLPS in heterochromatin compaction is not mutually exclusive. Within chromatin globules initially collapsed via PPPS, LLPS can be triggered due to the high concentration of proteins (H1, HP1, etc.) possessing IDRs that are capable of establishing weak multivalent interactions (Figure 2).

Whatever the mechanism of heterochromatin domain formation (LLPS, PPPS, or both to various extents), it is likely to be controlled by epigenetic modifications of histone tails. Of note, nucleosome arrays lacking these tails do not produce liquid droplets. Acetylation of histone tails in pre-formed droplets causes dissolving of droplets [44]. Bridging of chromatin fiber by HP1 depends on the presence of H3K9me3 modification [86] due to the presence of H3K9me3-binding chromodomain in HP1.

### 3.3. Role of Phase Separation in Active Chromatin Compartment Assembly

Analysis of Hi-C maps has demonstrated that spatial interactions between remote genomic elements exist preferentially within A and B chromatin compartments but to a much lesser extent between these compartments [52].

Active chromatin is less densely packed as compared to heterochromatin. Within the A chromatin compartment, there are no general organizers, such as HP1 or Polycomb proteins. However, long-distance interactions within the A compartment are common and, furthermore, in some cells, they are more pronounced than long-distance interactions within the B compartment [87]. Most of the long-distance interactions within the A compartment are related to the realization of various functional processes. Some of these spatial interactions represent enhancer-promoter loops joining phase-separated activating compartments assembled on promoters and enhancers. We shall address these regulatory interactions in the next section of this review. Here, we focus on shaping of the 3D genome mediated by the interaction of chromatin fiber with various nuclear bodies generated via LLPS. The most significant impact on the 3D genome is the recruitment of active genes to the sheared locations of transcription (transcription factories) and also to splicing speckles. The clustering of active RNA polymerases in molecular assemblies termed transcription factories is well documented (reviewed in [88,89,90]. However, the nature of transcription factories is still poorly understood. Some data suggest that RNA polymerase clusters exist in the absence of transcription, and that, to be transcribed, genes should be somehow recruited to these clusters. Another model proposes that initiated or elongating RNA polymerases are stochastically clustered (reviewed in [90]). Recent evidence suggests that the clustering of transcription complexes is regulated by phosphorylation of the RNA pol II C-terminal domain [18]. It is not clear whether there is any specificity in gene recruitment to the same transcription factory. Some studies demonstrate that closely located genes are assembled in transcription factories independently of their tissue specificity [91,92]; another study has provided evidence for the existence of tissue-specific transcription factories [93]. Of note, there are well-documented cases when remote genes, including genes located on different chromosomes, are transcribed in the same transcription factory [92,93]. Clearly, recruitment of genes to sheared transcription factories should be considered as an important factor of spatial genome organization.

Besides transcriptional factories, nuclear speckles also represent LLPS-derived compartments that attract active genes [38,94,95]. Although initial studies provided controversial results concerning the localization of active genes with respect to nuclear speckles [49,96], the more recent data strongly support the idea that nuclear speckles mediate spatial organization of the active chromatin compartment [94,97,98].

The list of liquid nuclear compartments to which various genes may be recruited is not limited by transcription factories and nuclear speckles. A subset of genes is attracted to Cajal bodies [99,100], and yet another subset to PML bodies [101,102,103]. All these interactions shape the 3D genome providing links between remote regions of a folded chromatin fiber. A sufficient number of such links may trigger PPPS.

## 4. Regulation of Gene Expression

### 4.1. Enhancers, Promoters and Enhancer-Promoter Communication

An important feature of the transcription control system in higher eukaryotes is the presence of remote enhancers that can be located hundreds of kilobases away from target promoters [104,105]. The mechanism of enhancers action is not fully understood. The most popular current model postulates that transcription factors and components of the transcription machinery attracted to enhancers interact with each other triggering LLPS, which results in the formation of a liquid activator compartment [27,106,107]. Indeed, Pol II, Mediator, and a number of known transcription factors possess IDRs [18,19,108,109] that can interact with each other triggering LLPS. Recent evidence suggests that enhancer RNA (eRNA) also contributes to the assembly of activating compartments on enhancers [110,111]. Superenhancers are typically composed of several enhancer blocks. Activating compartments associated with each of these blocks can fuse, giving rise to a common activating compartment (Figure 3) [112]. To be activated by an enhancer, a gene should be located within the above-described compartment. In the case of remote enhancers, this location becomes possible via looping out of an intervening DNA segment [113,114]. Clearly, transcription factors and components of the transcription machinery are also recruited to promoters where they form a phase-separated liquid compartment [18,107]. For some transcription factors, it was demonstrated that the activation of transcription by these factors is directly related to their ability to form phase-separated liquid condensates to which RNA polymerase II, Mediator, and other components of the transcription apparatus become attracted [115]. Furthermore, it has been shown that the ability of transcription factor TAZ, one of the downstream targets of the Hippo signaling pathway, to form phase-separated condensates is inhibited by Hippo signaling through LATS/NDR kinase-mediated phosphorylation [115].

Being brought to spatial proximity, liquid compartments associated with an enhancer and a promoter fuse, giving rise to a common compartment rich in proteins necessary for effective transcription initiation (Figure 3) [107,116]. Of note, this system allows for the assembly of various multicomponent complexes containing several enhancers and promoters. The existence of such complexes was first deduced based on 3C data [117,118] and was then demonstrated using a GAM protocol for studying 3D genome organization [119]. Notably, in our previously published study [120], we have proposed the model of enhancer-promoter communication within the so-called active chromatin microcompartment (ACM) representing a small volume inside the chromatin mash where enhancer and promoter are located in close proximity to each other but do not necessarily establish stable, direct contact. It is relevant to assume that ACM could originate as a result of the fusion of liquid droplets containing an enhancer and promoter.

### 4.2. Promoter Clearance

A characteristic feature of many eukaryotic promoters is the presence of paused Pol II elongation complexes [121]. The regulated release of these paused Pol II complexes allows for the synchronous activation of a number of silent genes in response to various stimuli [122,123]. The release of Pol II from promoter-proximal pausing is controlled by positive transcription elongation factor b (P-TEFb), a heterodimer of the kinase CDK9 and CCNT1 [124]. In cells, most of the P-TEFb is sequestered in the catalytically inactive HEXIM1/2-containing complex [125,126], whereas the active form of P-TEFb constitutes a part of a super elongation complex (SEC) [127,128]. A recent study demonstrates that the SEC complex forms nuclear puncta (foci) via LLPS directed by multivalent interactions of IDRs present in SEC subunits ELL and AFF4 [129]. The CDK9 subunit of P-TEFb also possesses IDR but is unable to form phase-separated condensates by itself. However, it can be adsorbed by phase-separated condensates formed by ELL and AFF. Finally, various functional tests demonstrated that the formation of SEC condensates at promoters is essential for the SEC-mediated release of paused Pol II [129]. Thus, LLPS is likely to play an important role in the control of paused promoter activity.

### 4.3. Transcription Elongation

Recent observations suggest that distinct liquid condensates are formed at gene bodies in the course of transcription [130,131]. The formation of these condensates may be promoted by the interaction of nascent RNA with a certain set of proteins, including the splicing machinery components [131]. Of note, switching of Pol II between different types of condensates is likely to be regulated by specific patterns of CTD phosphorylation [131,132]. The suppression of CTD phosphorylation resulted in a reduction in the occupancy of multiple splicing factor condensate components [131,132]. This observation implies that phosphorylated CTD serves as a scaffold for the assembly of transcription elongation-related phase-separated protein complexes. The relation of these complexes to the SC35 speckles is to be determined. The speckles are commonly regarded as locations of splicing components’ storage [133]. Yet, active genes tend to be located close to speckles [97], and some observations suggest that the spliceosome assembly may occur at the surface of speckles [134,135]. It should also be noted that in the above-discussed study [131], the assembly of mediator- and splicing factor-related condensates on actively transcribed genes controlled by superenhancers was characterized. Further research is necessary to determine whether these observations reflect the patterns of Pol II compartmentalization upon transcription of other types of genes.

## 5. Dysregulation of LLPS in the Cell Nucleus as a Driver of Pathology

The LLPS process being determined by the structure of cellular proteins and RNAs is affected by genetic mutations in a number of severe diseases, such as neurodegenerative disorders [136] and cancers [137]. These mutations cause various alterations in the structure of macromolecules (first of all, the number and affinity of their interacting domains) that influence their propensity to establish weak multivalent interactions with each other and, consequently, to nucleate and form stable condensates. In a cell nucleus, many DNA- and chromatin-interacting proteins are involved in the liquid droplet formation that facilitates their functions [138]. For instance, the IDR-containing ENL protein (a reader of acetylated histones with YEATS domain) responsible for the maintenance of an oncogenic state in leukemia self-associate at moderate level in normal conditions, promoting transcriptional elongation at target genes. In Wilms tumor (the most widespread pediatric kidney cancer), short deletion of the PP amino-acid motif and insertion of the NHL motif results in markedly increased self-association of the ENL in nuclear foci accompanied with upregulation of the ENL-bound promoters [139]. Strikingly, these pathological mutations influencing condensate formation occur not in IDR (as one would expect) but in the structured YEATS domain. This finding is potentially explained by the altered structure of the mutated domain that could be partially unfolded and, thus, mimic the properties of IDRs. Moreover, mutants with the deletion of PP and the insertion of the NHL amino-acid motifs demonstrated the same enhanced ability for condensate formation. Hence, in the case of abnormal LLPS in the nucleus, similar to other genetically-driven misfunctions, different genetic backgrounds lead to the appearance of the same pathological phenotype (aberrant condensate formation by the mutated protein).

The same is true for Rett syndrome (RTT), which is a postnatal neurodevelopmental disorder characterized by mental disability and autism-like symptoms. In RTT, different mutations in the C-terminal IDR and structured methyl-binding domain of the methyl CpG binding protein 2 (MeCP2) suppress the ability of this protein to form condensates in vivo [140]. In particular, the R168X mutant displays genome-wide transcriptional dysregulation, including the loss of repetitive element silencing and gross alterations of chromatin structure manifested in a reduced formation of HP1-alpha foci. MeCP2 is a key component of heterochromatin. Thus, it is relevant to assume that the disruption of MeCP2 condensates in RTT results in the decondensation of heterochromatin domains and most likely to changes in its epigenetic properties that lead to a disturbance of transcription repression of genes that are silent in the wild-type genotype.

The expansion of short nucleotide repeats is another example of genetic deficits affecting LLPS. It has been shown that CAG multiplication, which is characteristic of Huntington’s disease and spinocerebellar ataxias, promotes RNA aggregation in vivo and in vitro [141]. CAG-containing RNA formed condensates only in the following conditions: droplet formation occurred only with >30 triplet repeats, required Mg2+, and was inhibited by antisense oligonucleotide. These findings suggest that the formation of disease-associated RNA droplets requires high valency of interacting molecules, involves electrostatic interactions besides base pairing, and is sequence specific interacting molecules, involves electrostatic interactions besides base pairing and is sequence-specific.

## 6. Conclusions and Perspectives

The importance of simple physicochemical processes in the organization of living systems has long been known. A good example is the formation of membranes, which is guided by simple physicochemical processes. The role of such processes in the formation of non-membranous structures became apparent only relatively recently after the demonstration of the importance of forces arising under conditions of macromolecular crowding for maintaining the integrity of various structures in the cell nucleus. Separation of liquid phases is another physicochemical phenomenon that plays an extremely important role in the functioning of living systems due to the fact that it is this phenomenon that underlies the compartmentalization of the intracellular space. Compartmentalization is one of the fundamental characteristics of a living cell. Without intention to discuss various theories of the origin of life, we only note that all of these theories postulate the separation of a certain protocell from the surrounding inanimate matter. It is also easy to see that cell compartmentalization became more complicated with evolution. Functional compartmentalization of the eukaryotic cell nucleus is interconnected with the spatial genome organization and underlies the realization of genome activities such as replication and transcription. Various regulatory events, such as establishing enhancer-promoter communication, also rely on dynamic compartmentalization manifested by the assembly of activating compartments via LLPS. To this end, it is not surprising that compromising LLPS results in the development of various diseases, including cancer. The ambitious task is, therefore, to learn how one can modify and control LLPS within living cells. Of course, this control must be targeted. As discussed above, in many cases, the macromolecules undergoing LLPS are gathered on a certain platform, which may be protein, RNA, or even DNA. The clearest method to disrupt liquid condensates of a particular type is, thus, to target this platform. The approach would depend on the nature and stability of a platform, the number of interacting molecules, and the properties of interacting sites. One solution is to suppress the synthesis of a platform (if the platform is RNA or protein). However, the possibility of blocking interacting sites with some drugs may also be considered. Whatever solution is found, we may expect that drugs affecting the integrity of macromolecular condensates will eventually be developed. The cell nucleus is an equilibrium system stabilized by various interactions of different nature.

## Figures and Tables

**Figure 1 cells-09-02480-f001:**
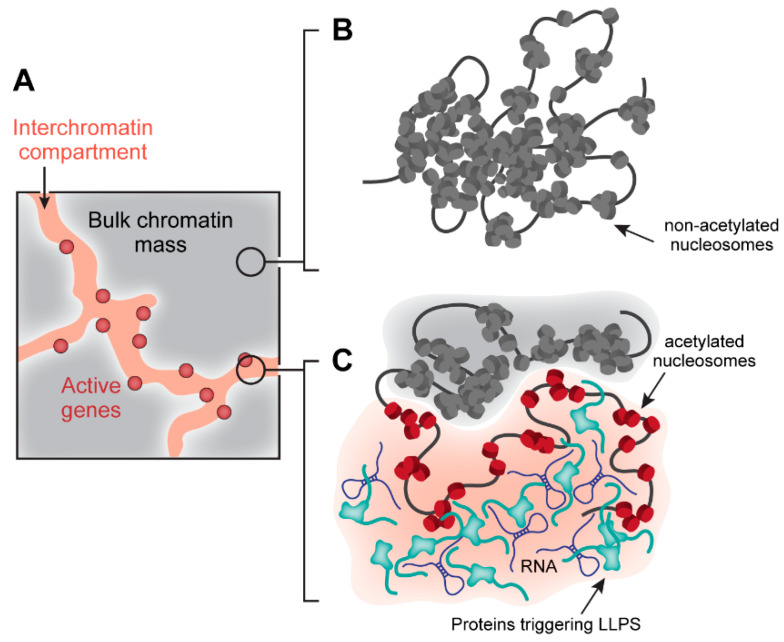
Interchromatin compartment (IC) and bulk chromatin mass form separate phases demarcated with actively transcribed genes lining channels of the IC (**A**). Phase separation is driven by interactions between non-acetylated nucleosomes (grey) in repressed chromatin (**B**) and weak multivalent interactions between IDR-containing proteins (light blue), RNA, and acetylated chromatin (red) in IC channels (**C**).

**Figure 2 cells-09-02480-f002:**
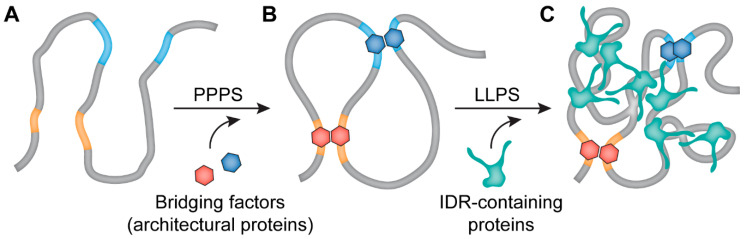
A putative two-step process of the chromatin droplet formation. At the first step, architectural proteins (red and blue hexagons), which can interact with each other, bind to its motifs in the DNA (**A**). This process results in the formation of loops (**B**) and partial condensation of the chromatin chain (polymer-polymer phase separation, PPPS). Next, IDR-containing chromatin-binding proteins establish multiple weak interactions with each other and with other components of chromatin collapsing the chromatin chain into a droplet (liquid-liquid phase separation; **C**).

**Figure 3 cells-09-02480-f003:**
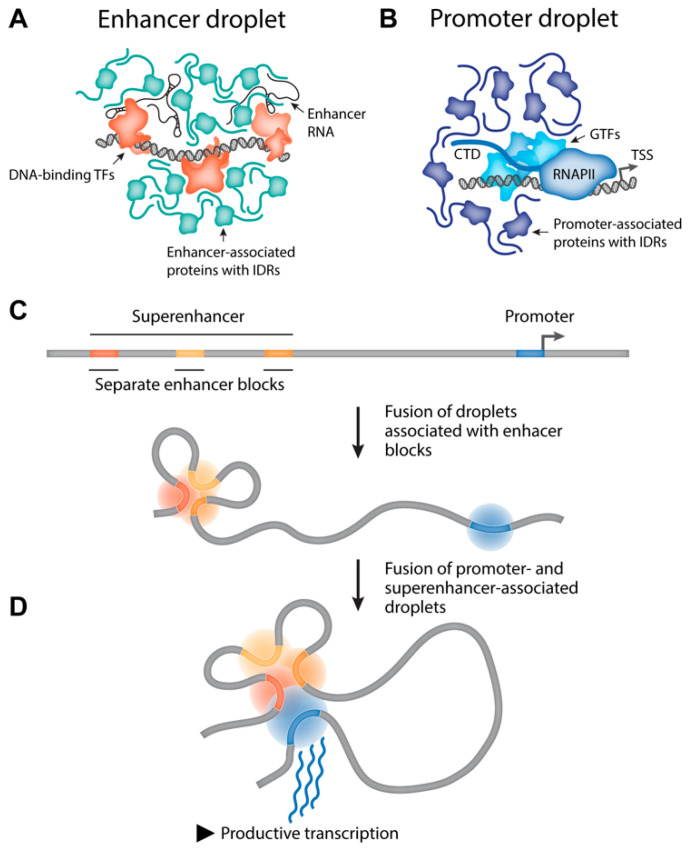
Enhancer-promoter contacts may be formed by the LLPS. Multivalent interactions between transcriptional machinery components drive the formation of liquid droplets at enhancer blocks within the superenhancer (**A**) and at the controlled promoter (**B**). Distinct droplets at the superenhancer may be fused due to being located in close proximity to each other, giving rise to a “superdroplet” encompassing the entire superenhancer region (**C**). 1D- or 3D-scanning of the nuclear space results in superpositioning of the superenhancer and promoter that leads to fusion of their liquid droplets (**D**). This structure, characterized by high local concentration of transcriptional activators, facilitates the productive transcription initiation from the promoter.

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
