# Peer review of "Divide and Rule: Phase Separation in Eukaryotic Genome Functioning"

_cells, 2020, doi:10.3390/cells9112480_

Round 1

Reviewer 1 Report

Razin & Ulianov summaries in their review “ Divide and rule: phase separation in the functioning 2 of eukaryotic genome“ the current knowledge of how liquid-liquid and polymer-polymer phase separation contribute to organization within the nucleus. The review covers the role of phase separation in genome organization (chromatin territories’, heterochromatin organization etc.) and transcriptional regulation (enhancer and promotor organization, promotor clearance, transcriptional elongation etc.). It is informative and comprehensive and will be a worthwhile contribution. However, the language is difficult to read. I strongly recommend help of a English native speaker also for a Grammar check. Specifically, sentences should be shortened.

Specific points:

Please define “gelation” starting from line 149.

Author Response

Following the reviewer’s suggestion, we performed a Grammar check and revised the overall writing.

In the revised version of the MS, we defined “gelation” as follows: “Gelation is a transition from a solution of dispersed monomers and oligomers to a system-spanning network [76]. This process, termed liquid–gel phase separation (LGPS), differs from PPPS described below (see [77])”.

Reviewer 2 Report

In this review Razin and Ulianov introduce the basic knowledge about the role of  liquid-liquid phase separation in genome organization. This is an interesting and "hot" topic.

General Comment: The authors should improve the writing of the review. In the text there are many convoluted sentences that significantly affect the clarity of the message.

The introduction can be improved. It appears too short and condensed.

A comment about possible caveats of the in vitro studies on liquid-liquid phase separation is demanded.

Author Response

Following the reviewer’s suggestion, we substantially improved the writing of the review

In the revised version of the MS, we also improved the Introduction by adding the following parts:

  1. The eukaryotic cell nucleus contains several functional compartments, such as the nucleolus, speckles, Cajal bodies, and PML bodies [1]. These compartments are not surrounded by membranes but yet are clearly distinct from the rest of the nucleoplasm. The mechanisms of nuclear compartments assembly have been discussed over many years [2–5]. Another actively discussed issue concerns the relationship between cell nucleus functional compartmentalization and genome spatial organization [1, 6, 7]”.
  2. To this end, it should be mentioned that a rapid exchange with the nucleoplasm is typical for proteins deposited in various nuclear compartments [31, 32]”.
  3. Below, we will discuss how various phase separation mechanisms contribute to eukaryotic cell nucleus compartmentalization and 3D organization of the genome. The role of phase separation in the assembly of nuclear bodies has been addressed in several recent reviews [37–39]. Therefore, here we focus on the recent data demonstrating the contribution of LLPS and PPPS in spatial genome organization, assembly of active and repressed chromatin compartments, and transcription control. We discuss the current model of enhancer action that postulates an LLPS-driven assembly of activating compartments on enhancers and superenhancers. Finally, we consider human pathologies caused by LLPS deregulation”.

We also improved the part of the MS where in vitro studies of the LLPS are addressed: “It should be mentioned, however, that conditions in the cell nucleus differ substantially from those in vitro. In particular, in the above-described experiments, the level of molecular crowding was not comparable to that within the cell nucleus. Further, the presence of high amounts of various RNA molecules can seriously affect protein interactions. Hence, both the ability and inability of a specific protein to form liquid condensates in vitro should be treated with caution when the potential contribution of this protein in the formation of liquid condensates within the cell nucleus is considered”.

Round 2

Reviewer 1 Report

all points of the reviewers' have been addressed. Especially, the English language and style has been strongly improved for a much better readability.